# Cereblon Deficiency Contributes to the Development of Elastase-Induced Emphysema by Enhancing NF-κB Activation

**DOI:** 10.3390/antiox11101980

**Published:** 2022-10-04

**Authors:** Eun-Young Heo, Kyoung-Hee Lee, Jisu Woo, Jiyeon Kim, Chang-Hoon Lee, Kyung-Jin Lee, Yun-Kyu Kim, Chul-Gyu Yoo

**Affiliations:** 1Department of Internal Medicine, SMG-SNU Boramae Medical Center, Seoul 07061, Korea; 2Division of Pulmonary and Critical Care Medicine, Department of Internal Medicine, Seoul National University Hospital, Seoul 03080, Korea; 3Department of Internal Medicine, Seoul National University College of Medicine, Seoul 03080, Korea; 4Department of Convergence Medicine, Asan Institute for Life Sciences, University of Ulsan College of Medicine, Asan Medical Center, Seoul 05505, Korea

**Keywords:** cereblon, emphysema, lung inflammation, NF-κB

## Abstract

Cereblon (CRBN) has been shown to play an essential role in regulating inflammatory response and endoplasmic reticulum stress, thus mediating the development of various diseases. However, little is known about the roles of CRBN in chronic obstructive pulmonary disease (COPD) pathogenesis. We found that the protein levels of CRBN in lung homogenates from patients with COPD were lower than those from never smokers and smokers. The CRBN protein level was positively correlated with the forced expiratory volume in 1 s (FEV1)/forced vital capacity (FVC). To investigate the role of CRBN in modulating elastase-induced emphysema, we used *Crbn* knockout (KO) mice. Elastase-induced emphysematous changes were significantly aggravated in *Crbn* KO mice. Neutrophil infiltration, lung cell injury, and protein leakage into the bronchoalveolar space were more severe in *Crbn* KO mice than in wild-type (WT) mice. Furthermore, *Crbn* KO resulted in the elevated release of neutrophilic chemokines and inflammatory cytokines in lung epithelial cells and macrophages. The transcriptional activity of nuclear factor-κB (NF-κB) was significantly increased in *Crbn* knocked-down cells. In conclusion, *Crbn* deficiency might be involved in the development of emphysema by enhancing NF-κB activation, suggesting that targeting CRBN might be an effective therapeutic approach for the treatment of COPD.

## 1. Introduction

Chronic obstructive pulmonary disease (COPD) is a major healthcare problem worldwide. The Global Burden of Disease Study 2015 estimated the global prevalence of COPD at approximately 174 million cases, and COPD ranked third among fatal diseases with approximately 3.2 million patients dying of the disease [1]. Emphysema is a primary phenotype of COPD and is defined as the irreversible destruction of alveolar structures and enlargement of the airspaces [2]. Oxidant/antioxidant or protease/antiprotease imbalances are pivotal to the pathogenesis of emphysema [3]. Considerable efforts have been made over the past decade to find a novel therapeutic to slow the inflammation of the lung and the progression of emphysema [4]. However, there is no effective treatment to change the natural course of emphysema because of the complexity of inflammatory signaling and mechanisms [5].

Cereblon (CRBN) was initially discovered as a candidate gene for a mild form of autosomal recessive non-syndromic mental retardation, and it has since been reported as a direct molecular target for the teratogenicity of thalidomide and the cytotoxicity of immunomodulatory drugs [6,7]. CRBN has been extensively studied and found to regulate the large conductance calcium- and voltage-activated potassium (BK_Ca_) channels [8,9] and CLC-1 chloride channels [10], bind to immunomodulatory drugs, and participate in the death of hematologic cancer cells [11,12,13]. Furthermore, CRBN has lately attracted considerable attention for its effect on regulating inflammation. It has been reported that CRBN inhibits proinflammatory cytokine production by suppressing nuclear factor-κB (NF-κB) activation via the attenuation of tumor necrosis factor (TNF) receptor-associated factor 6 (TRAF6) and ubiquitination of TAK1-binding protein 2 (TAB2) [14]. In addition, CRBN is known to inhibit the lipopolysaccharide (LPS)-induced production of inflammatory cytokines by mediating the ubiquitination and degradation of c-Jun [15]. It also binds to adenosine monophosphate-activated protein kinase (AMPK) and inhibits the activation of AMPK, which plays a vital role in regulating metabolic processes, the inflammatory response, oxidative stress, and endoplasmic reticulum stress, thus mediating the development of various diseases [6,16,17]. However, whether CRBN affects lung emphysema, and the underlying mechanisms of emphysema pathogenesis are not known. The objective of this study was to investigate the role of CRBN in the development of elastase-induced emphysema in mice and the associated molecular mechanisms.

## 2. Materials and Methods

### 2.1. Mice

Female wild-type (WT) BALB/c mice were purchased from Koatech Laboratory Animal Company (Pyeongtaek, Korea), and *Crbn* knockout (KO) mice in a BALB/c genetic background were kindly donated by Dr. Kyung Jin Lee (Asan Medical Center, Seoul, Korea). All mice used were sex-matched at 6–8 weeks of age (20–22 g) and housed in the animal facility of Seoul National University Hospital under specific pathogen-free barrier conditions. Animal experiments were approved by the Institutional Animal Care and Use Committee (number 19-0220-S1A1(1)) of Seoul National University Hospital.

### 2.2. Intratracheal Instillation of Elastase

Mice were anesthetized and given 0.5 units of porcine pancreas elastase (Sigma-Aldrich, St. Louis, MO, USA) in 100 μL of saline or saline alone via intratracheal injection. Mice were sacrificed on day 2 and day 14 after elastase instillation.

### 2.3. Analysis of Bronchoalveolar Lavage Fluid (BALF)

On days 2 and 14 after elastase administration, the lungs of terminally anesthetized mice were lavaged with 1 mL of cold phosphate-buffered saline (PBS). BALF was centrifuged at 1500 rpm at 4 °C for 10 min and the supernatants were then collected for the measurement of the levels of lactate dehydrogenase (LDH) and protein. The total cell count was determined with a hemocytometer and BAL cell distribution was quantified in cytospin preparations after staining with Diff-Quik dye (Sysmex, Kobe, Japan).

### 2.4. Emphysema Measurement and Immunohistochemistry

The right lungs were fixed in 4% paraformaldehyde solution (USB products, USA) for 24 h, embedded in paraffin, and sectioned in 4 µm-thick slices. The slides were stained with hematoxylin and eosin (H&E). Four randomly selected ×100 fields per specimen were photographed in a blinded manner. Emphysema was quantified by measuring the mean linear intercept (MLI). The MLI was measured by placing four 1000-μm lines over each field. The total length of each line divided by the number of alveolar intercepts gives the average distance. The non-parenchymal area was not included. Lung tissues were placed on slides using the Discovery XT automated immunohistochemistry stainer (Ventana Medical Systems, Inc., Tucson, AZ, USA). Tissue sections were deparaffinized and rehydrated. Cell conditioning 1 standard (pH 8.4 buffer containing Tris/borate/EDTA) was used for antigen retrieval. The sections were incubated with anti-8-hydroxy-2′-deoxyguanosine (8-OHdG), anti-4 hydroxynonenal (4HNE), or anti-matrix metallopeptidase 9 (MMP9) antibody for 32 min at 37 °C, washed, and incubated with a secondary antibody for 20 min at 37 °C. After successive washes, slides were incubated with 3, 3-diaminobenzidine (DAB) H_2_O_2_ substrate for 8 min at 37 °C, followed by a hematoxylin and bluing reagent counterstain. Stained cells were observed under a microscope (EVOS XL Core Cell Imaging System, Thermo Fisher Scientific, Waltham, MA, USA.).

### 2.5. Cells and Reagents

Normal human bronchial epithelial cells (BEAS-2B) were maintained in a defined keratinocyte serum-free medium (Gibco by Life Technologies, Grand Island, NY, USA) at 37 °C under 5% CO_2_. Bone marrow-derived macrophages (BMDMs) were obtained from bone marrow by flushing the tibia and femur of a BALB/c mouse. Bone marrow cells were cultured in DMEM supplemented with 20% FBS and 30% L929 cell supernatant for 5–7 days. BMDMs were resuspended in DMEM supplemented with 10% FBS and used in experiments. Human sputum neutrophil elastase (NE) was purchased from Elastin Products Co. (Owensville, MO, USA). NE was dissolved in a solution of 50% glycerol and 50% 0.02 M NaOAc (pH 5) (vehicle control, VC). Antibodies used for protein detection were anti-phospho-SAPK/JNK (p-SAPK/JNK) (Thr183/Tyr185), anti-p-p38 (Thr180/Tyr182), anti-total p38, anti-p-ERK (Thr202/Tyr204), anti-total ERK, and anti-IκBα antibodies (Cell Signaling Technology, Danvers, MA, USA); anti-CRBN antibody (Novus Biologicals, Centennial, CO, USA); anti-8-OHdG (Bioss antibodies Inc., Woburn, MA, USA); anti-total JNK and anti-GAPDH antibodies (Santa Cruz Biotechnology Inc., Santa Cruz, CA, USA); and anti-4HNE and anti-MMP9 antibodies (Abcam, Cambridge, MA, USA). Control siRNA and CRBN siRNA (Assay ID: s27634) were purchased from Thermo Fisher Scientific.

### 2.6. Preparation of Cigarette Smoke Extract (CSE)

Commercial cigarettes (THIS; KT&G Corp., Daejeon, South Korea) were smoked continuously by a bottle system connected to a vacuum machine (Gast Manufacturing Inc., Benton Harbor, MI, USA). The smoke from 20 cigarettes was bubbled in 60 mL of PBS. The large insoluble particles were removed by filtering the solution through a 0.22 μm filter.

### 2.7. Isolation of RNA and Quantitative Real-Time PCR

Total RNA was isolated using an RNeasy kit (Qiagen, Hilden, Germany). cDNA was synthesized using the Reverse Transcription System (Promega, Madison, WI, USA). Thr Power SYBR Green PCR Master Mix (Applied Biosystems, Carlsbad, CA, USA) was used for amplification. The primers used in the study were as follows: CRBN (fwd: 5′-AGC ATG GTG CGG AAC TTA ATC-3′, rev: 5′-ATC TCT GCT GTT GTC CCA AAC-3′), KC (fwd: 5′-TGT CAG TGC CTG CAG ACC AT-3′, rev: 5′-CCT GAG GGC AAC ACC TTC A-3′), TNF-α (fwd: 5′-CAC AGA AAG CAT GAT CCG CGA CGT-3′, rev: 5′-CGG CAG AGA GGA GGT TGA CTT TCT-3′), IL-6 (fwd: 5′-AAC GAT GAT GCA CTT GCA GA-3′, rev: 5′-GAG CAT TGG AAA TTG GGG TA-3′), IL-1β (fwd: 5′-CCA GCT TCA AAT CTC ACA GCA G-3′, rev: 5′-CTT CTT TGG GTA TTG CTT GGG ATC-3′), GAPDH (fwd: 5′-ACG GCA AAT TCA ACG GCA CAG-3′, rev: 5′-TGG GGG CAT CGG CAG AAG G-3′).

### 2.8. Measurement of Cytokines

The levels of interleukin-8 (IL-8), keratinocytes-derived chemokine (KC), macrophage inflammatory protein 2 (MIP2), tumor growth factor-α (TNF-α), and IL-6 in cell culture supernatants or in BALF were measured using a commercially available DuoSet ELISA kit (R&D System, Minneapolis, MN, USA). The concentrations of TNF-α, IL-6, and IL-1β in BALF were measured using a Bio-Plex Pro™ cytokine assay kit (Bio-Rad, Hercules, CA, USA) according to the manufacturer’s instructions.

### 2.9. Protein Extraction and Western Blot Analysis

Total cellular extracts were prepared using 1× Cell Lysis Buffer (Cell Signaling Technology) supplemented with 1 mM PMSF. Frozen lung tissues (SNUH IRB Number:H-1309-073-521) were homogenized in tissue extraction buffer (Life Technologies) containing a protease inhibitor mixture (Sigma) and phosphatase inhibitor mixture (Sigma). Protein concentration was determined using the Bradford assay (Bio-Rad, Hercules, CA, USA). Cell extracts were subjected to sodium dodecyl sulfate–polyacrylamide gel electrophoresis (SDS-PAGE), and gels were transferred to Hybond ECL nitrocellulose membranes (Thermo Fisher Scientific) for 100 min at 90 V. The membranes were blocked with 5% skim milk in 1× Tris-buffered saline (TBS) containing 0.1% Tween 20 (TBS-T) for 1 h at room temperature. After successful washes, the membranes were incubated with horseradish peroxidase (HRP)-conjugated secondary antibodies for 1 h. Blots were developed using a West Pico Western blot detection kit (Thermo Fisher Scientific).

### 2.10. Luciferase Assay

Cells were transfected with the NF-κB reporter plasmid or control plasmid using a Neon Transfection System (Thermo Fisher Scientific) according to the manufacturer’s specifications. Luciferase activity was determined using a luciferase assay kit (Promega, Madison, WI, USA).

### 2.11. Statistical Analysis

Data were analyzed using Prism (version 5; GraphPad Software, San Diego, CA, USA). Differences between means were explored using the Kruskal–Wallis test via the Dunn’s multiple comparison post hoc test. To evaluate correlations in the human samples, the Pearson correlation coefficient was calculated. Differences between groups were evaluated by Student’s *t*-test. A *p*-value of <0.05 was considered significant.

## 3. Results

### 3.1. The Expression Level of CRBN Was Decreased in Lung Tissues of Patients with COPD and Correlated with the Forced Expiratory Volume in 1 s (FEV1)/Forced Vital Capacity (FVC) Ratio

To investigate the functional role of CRBN in emphysema development in the human lung, we first measured CRBN expression in surgically resected human lung tissues with immunoblotting. The expression of CRBN protein was lower in COPD lungs (n = 8) than in non-COPD lungs (never smokers; n = 8, smokers; n = 8) (Figure 1a,b). CRBN expression was not significantly different between never smokers and smokers (Figure 1a,b). Cigarette smoking is the main risk factor for COPD, yet only 20–30% of smokers develop COPD [18]. These findings suggest that CRBN plays a role in the development of COPD regardless of smoking. We further investigated the association between clinical parameters and the level of CRBN expression. The clinical parameters included age, body mass index, pack-years of smoking (py), pre-bronchodilator FEV1, the ratio of FEV1 to FVC, diffusion capacity (DLco), COPD assessment test (CAT) score, St. George’s Respiratory Questionnaire (SGRQ), and Modified Medical Research Council scale (mMRC). Among these clinical parameters, only FEV1/FVC was significantly correlated with the level of CRBN expression (Figure 1c–k). A FEV1/FVC ratio of less than 70% indicated airflow limitation and the possibility of COPD. The correlation between the ratio of FEV1/FVC and the expression of CRBN showed that a decreased level of CRBN is consistently associated with COPD development.

### 3.2. Crbn KO Exaggerated Elastase-Induced Emphysema in Mice

Rodent models of elastase-induced emphysema were established [19]. To evaluate whether the expression of CRBN affected the development of elastase-induced emphysema in mice, we treated WT and *Crbn* KO BALB/c mice with elastase intratracheally. *Crbn* KO was confirmed via RT-PCR analysis (Figure 2a). On day 14 after intratracheal elastase administration, more prominent alveolar destruction and airspace enlargement were observed in elastase-treated mice than in saline-treated mice. These changes were more exaggerated in *Crbn* KO mice than in WT mice (Figure 2b). Consistent with the morphological results, the MLI (a measurement of the mean interalveolar septal wall distance) was markedly higher in the lungs of elastase-treated *Crbn* KO mice than in the lungs of WT mice (Figure 2c).

### 3.3. Crbn KO Increased Elastase-Induced Inflammation and Cellular Injury

The lungs of elastase-treated mice showed marked inflammation with immune cell infiltration on day 2 that returned to basal levels on day 14. Elastase-induced inflammation and emphysema were more prominent in *Crbn* KO mice than in WT mice (Figure 3a). To investigate whether elastase-induced inflammation mimicked those of patients with COPD, we analyzed bronchoalveolar lavage fluid (BALF). Neutrophils are the most abundant inflammatory cells in the bronchi of patients with COPD, and increased neutrophilic inflammation is characteristic of acute exacerbation of COPD [20]. The total number of BALF cells was significantly higher on day 2 after elastase administration in *Crbn* KO mice than in WT mice (Figure 3b). Along with the total cell count, neutrophils were markedly increased in BALF from elastase-treated *Crbn* KO mice (Figure 3c). However, most of the inflammatory cells in the BALF disappeared 14 days after elastase administration, when the emphysematous changes were established. The concentration of LDH (Figure 3d) and protein (Figure 3e) in the BALF were significantly higher in elastase-treated *Crbn* KO mice than in elastase-treated WT mice. On day 14 after elastase administration, the levels of LDH and protein in BALF were reduced to the baseline in the same manner as the neutrophils. We measured the levels of neutrophilic chemokines and inflammatory cytokines: KC, TNF-α, IL-6, and IL-1β in total lung tissues (Figure 3f–i) and BALF (3j-m). The expression levels of KC and TNF-α mRNA were significantly higher on day 2 after elastase administration in *Crbn* KO mice than in WT mice (Figure 3f,g). Only IL-6 in BALF was higher in elastase-treated *Crbn* KO mice than in elastase-treated WT mice (Figure 3l). These findings suggested that neutrophilic inflammation and cellular injury precede emphysema development, and CRBN deficiency could contribute to elastase-induced neutrophilic inflammation, leading to cellular injury.

### 3.4. Crbn KO Enhanced Production of Inflammatory Cytokines and Chemokines by Increasing NF-κB Activation in Lung Epithelial Cells and Macrophages

Interleukin-8 (IL-8) is a neutrophil chemoattractant whose levels in sputum samples of patients with COPD are increased [21]. To investigate the role of CRBN on IL-8 production, we transiently transfected BEAS-2B cells with the control and *Crbn* siRNAs and treated the cells with NE (1 U/mL) or CSE (1%) for 24 h. NE or CSE treatment induced IL-8 production in BEAS-2B cells. NE or CSE-induced IL-8 production was higher in *Crbn* siRNA-transfected cells than in the control siRNA-transfected cells (Figure 4a,b).

Because alveolar macrophages play a central role in modulating inflammation by producing many inflammatory proteins in patients with COPD [22], we next investigated the effect of *Crbn* KO not only in bronchial epithelial cells, but also in macrophages. BMDMs from WT and *Crbn* KO mice were treated with NE (1 U/mL), CSE (1%), or LPS (100 ng/mL). We measured the concentrations of KC, MIP2, TNF-α, and IL-6 in the supernatant of BMDMs. In NE-treated BMDMs, the concentrations of KC, MIP2, TNF-α, and IL-6 were not significantly different between WT and *Crbn* KO mice (Figure 4c–f). In CSE-treated BMDMs, KC and MIP2 were significantly higher in *Crbn* KO cells than in WT cells (Figure 4g,h). However, CSE induced neither TNF-α nor IL-6 (Figure 4i,j). In LPS-treated BMDMs, the production of KC, MIP2, TNF-α, and IL-6 was significantly higher in *Crbn* KO cells than in WT cells (Figure 4k–n).

Many inflammatory cytokines/chemokines are regulated by the mitogen-activated protein kinases (MAPKs) and NF-κB pathways [23]. Given the increased neutrophilic inflammation in *Crbn* KO mice and the increased release of inflammatory cytokines/chemokines in *Crbn*-deficient cells, we investigated the effect of decreased levels of CRBN on the activation of MAPKs and NF-κB. The activation of the MAPK pathway was determined with the Western blot analysis of phosphorylated MAPKs (ERK, JNK, and p38). The NF-κB activation process involves IκBα phosphorylation by IκB kinases (IKKs), and the subsequent degradation of IκBα, and nuclear translocation of NF-κB [23]. Therefore, activation of the NF-κB pathway was determined with a Western blot analysis of IκBα and luciferase activity assay in cells transfected with an NF-κB-luciferase reporter construct. Both NE and CSE induced IκBα degradation and MAPK activation, which were not affected by *Crbn* deficiency (Figure 5a,b, Appendix A). LPS-induced degradation of IκBα also was not changed by *Crbn* KO (Figure 5c). However, NF-κB transcriptional activity was significantly increased in *Crbn* knocked-down cells treated with CSE (Figure 5d). These results suggest that exaggerated inflammation in *Crbn* KO mice and cells might be due to increased activation of the NF-κB pathway, probably following IκBα degradation.

### 3.5. Crbn KO Increased Elastase-Induced Oxidative Damage and MMP9 Expression

Oxidative stress is recognized as an important predisposing factor in COPD pathogenesis, [24] and MMPs are known to contribute to elastin and collagen matrix degradation, resulting in emphysema [25]. To elucidate whether *Crbn* KO affects elastase-induced oxidative damage, we performed immunohistochemical staining for 8-OHdG (one of the major products of DNA oxidation) and 4HNE (a product of lipid peroxidation) in lung tissues from elastase-instilled WT and *Crbn* KO mice. The intensity of 8-OHdG and 4HNE staining on day 2 after elastase instillation was greater in *Crbn* KO mice than in WT mice (Figure 6a–d). MMP-9 is one of the predominant elastolytic enzymes in patients with COPD [26]. Increased intensity of MMP9 staining was found in the lungs of elastase-treated *Crbn* KO mice, and the MMP9-positive cells were mostly recruited immune cells (Figure 7a,b). Furthermore, NE- or LPS-induced MMP9 expression was increased in *Crbn*-deficient lung epithelial cells and macrophages (Figure 7c,d). These results indicate that *Crbn* deficiency could also promote elastase-induced oxidative stress and proteinase expression.

## 4. Discussion

The role of CRBN in the pathogenesis of COPD has not been investigated. Our present findings show that decreased CRBN contributes to the development of emphysema by increasing neutrophilic inflammation, oxidative damage, and proteinase (MMP9) expression. One plausible mechanism of increased inflammation is that CRBN enhances the transcriptional activity of NF-κB in lung epithelial cells and macrophages.

Neutrophilic inflammation is a prominent feature in the COPD airway, and neutrophil products are thought to be key mediators causing emphysema [27]. Airway neutrophilia is associated with the rate of decline in lung function [28]. This is supported by the finding that the protein expression of CRBN in total lung tissue lysates from patients with COPD was lower than that in lung tissue lysates of both never smokers and smokers. The FEV1/FVC ratio, which is an essential parameter for COPD diagnosis, was significantly correlated with CRBN levels. Lung function parameter, especially FEV1/FVC, correlates with the extent of emphysema [29]. This suggests COPD patients with lower CRBN levels have more emphysematous lung. As expected, in the elastase-treated mouse model, an exaggerated emphysematous change was observed in *Crbn* KO mice compared to WT mice. More neutrophils were recruited to the lungs, and lung cell damage as indicated by LDH release and protein leakage into BALF was more severe in elastase-treated *Crbn* KO mice than in elastase-treated WT mice. Resident lung cells, such as epithelial cells and alveolar macrophages, play an essential role in initiating and modulating inflammation by producing a variety of inflammatory cytokines and chemokines in patients with COPD [22]. Therefore, we investigated the role of CRBN in inflammatory responses in both cell types.

Because cigarette smoking is the major risk factor for COPD, we stimulated both cell types with CSE as well as with elastase. Elastase or CSE-induced IL-8 production was significantly enhanced with *Crbn* knockdown in lung epithelial cells. Although elastase alone did not induce KC, MIP2, TNF-α, or IL-6 in WT or *Crbn* KO BMDMs, CSE-mediated secretion of neutrophilic chemokines such as KC and MIP2 was exaggerated in *Crbn* KO BMDMs. NE was reported to upregulate inflammatory cytokine production in macrophages differentiated from primary peripheral blood mononuclear cells [30]. Similar to our results, NE treatment alone did not increase the levels of TNF-α and IL-1β. However, in response to LPS treatment, NE enhanced the release of those cytokines [30]. In chronic inflammatory environments such as COPD, multiple factors interact to stimulate inflammatory responses. NE alone does not seem enough to induce an inflammatory response. Cigarette smoke (CS) and CS extracts (CSE) were known to induce inflammatory cytokines in several types of cells including lung epithelial cells and macrophages. However, in this study, CSE did not induce the release of proinflammatory cytokines, such as TNF-α and IL-6 in BMDM. In our previous study, we performed a series of experiments to explain the discrepancy in cytokine production by CSE [31]. As well as in BMDM, incubation with CSE did not increase the release of TNF-α, IL-6, and IL-1β in RAW264.7 cells, peritoneal macrophages, and bronchoalveolar lavage (BAL) macrophages. Co-treatment with CSE significantly suppressed the levels of TNF-α, IL-6, and IL-1β mRNA/protein in LPS-treated RAW264.7 cells, peritoneal macrophages, and BAL macrophages via p300 downregulation [31]. We do not know exactly why CSE did not increase inflammatory cytokine production unlike the findings of other researchers, but a non-standardized CSE extraction method or the use of other types of cigarettes might be the cause. Therefore, to clarify the role of CRBN on the production of inflammatory chemokines and cytokines in macrophage, we used LPS as a stimulus.

LPS is a toxic component of Gram-negative bacteria that provokes profound inflammation [32]. Patients with COPD with exacerbations have a significantly increased number of inflammatory cells in the sputum and an accelerating progression of emphysema compared with patients with COPD without exacerbations [33,34]. Most exacerbations are precipitated by viral or bacterial infections, and Gram-negative bacteria are the most commonly isolated pathogens in such cases [35]. Even a single intratracheal administration of LPS to mice with elastase-induced emphysema can provoke inflammation, increase protease production, and finally lead to more severe emphysematous change [36]. We also evaluated the effect of CRBN in LPS-treated WT and *Crbn* KO BMDMs. LPS strongly induced inflammatory chemokines/cytokines, which were remarkably enhanced in *Crbn* KO BMDMs. This finding suggests that CRBN may have a role in acute inflammatory response as well as chronic inflammation.

CSE-induced IL-8 production, and LPS-induced proinflammatory cytokines are known to be regulated by the MAPK and NF-κB pathways [37,38,39]. However, MAPK activation by elastase or CSE was not affected by CRBN expression. NF-κB activation involves IκBα phosphorylation by IKKs and the subsequent degradation of IκBα. NF-κB liberated from the IκB complex translocates into the nucleus, resulting in the expression of many inflammatory genes, including those encoding TNF-α and interleukins [37,39]. Although *Crbn* KO did not influence IκBα degradation in cells treated with elastase, CSE, or LPS in our study, the transcriptional activity of NF-κB was enhanced when the level of CRBN was reduced. In this regard, *Crbn* KO might increase neutrophilic inflammation and lead to profound emphysema by enhancing NF-*κ*B activation.

In addition to inflammation, oxidative stress, proteinase/anti-proteinase imbalance, activation of autophagy, cellular senescence, and apoptosis are also considered important pathogenic factors of COPD [40]. MMPs and the lipid peroxidation product 4HNE are reported to be elevated in lung tissues of patients with COPD [41]. Consistent with this, the lungs of elastase-treated mice contained elevated numbers of cells positive for 8-OHdG, 4HNE, and MMP9, an effect that was augmented with KO of *Crbn*. Because oxidative stress, proteinase activity, and inflammation are reciprocally regulated, CRBN appears to regulate each process either directly or indirectly. However, a reduced level of CRBN did not affect CSE-induced cellular aging, activation of autophagy, or apoptosis in lung epithelial cells (Appendix A). Class A scavenger receptors (SR-A) are mainly expressed in macrophages. SR-A could increase inflammation, [42] and COPD is associated with SR-A gene sequence variation [43]. However, *Crbn* KO did not change the levels of SR-As as in SR-A1 or the macrophage receptor with the collagenous structure (MARCO) in LPS- or elastase-treated BMDMs (Appendix A).

## 5. Conclusions

In conclusion, *Crbn* deficiency contributes to the development of emphysema by increasing neutrophilic inflammation, oxidative damage, and proteinase expression (Figure 8). This study provides new evidence for the therapeutic potential of targeting CRBN for the treatment of COPD/emphysema. However, the molecular mechanism of how CRBN regulates the transcriptional activity of NF-κB in lung epithelial cells and macrophages is unclear. To elucidate the mechanism, further detailed studies are required. Moreover, if the role of CRBN in acute inflammatory conditions is investigated, it would be helpful to regulate COPD exacerbation.

## Figures and Tables

**Figure 1 antioxidants-11-01980-f001:**
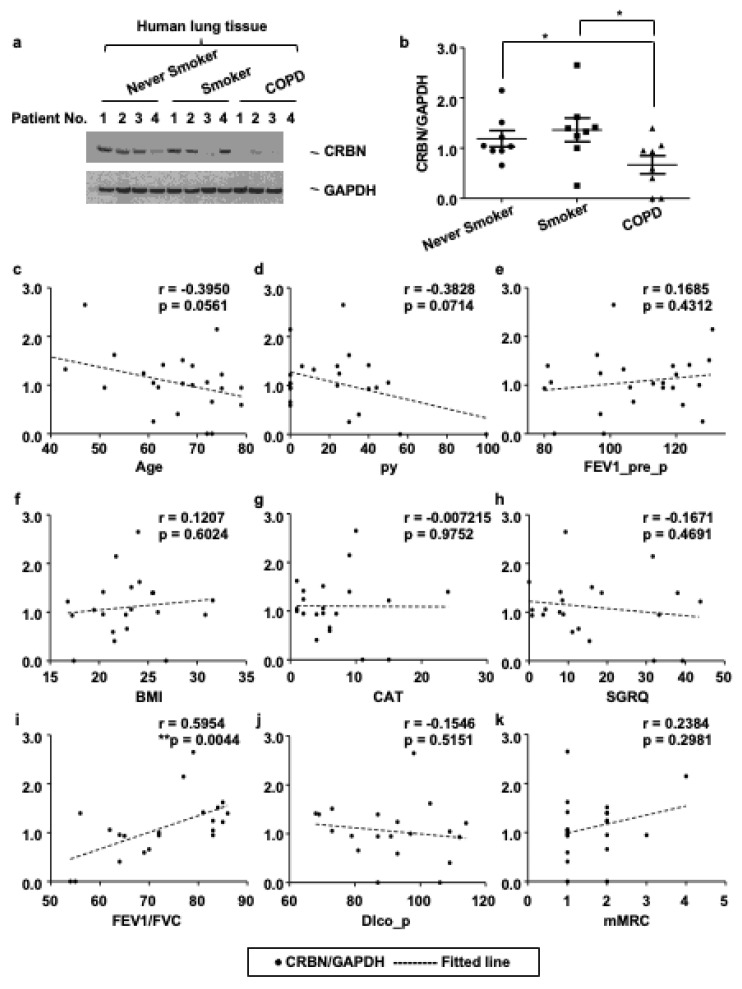
The level of CRBN expression was decreased in lung tissues of patients with COPD and correlated with the FEV1/FVC ratio. (**a**) Lung homogenates from never smokers (n = 8), smokers (n = 8), and patients with COPD (n = 8) were subjected to Western blot analysis for CRBN and GAPDH. (**b**) Gel data were quantified using Scion image densitometry. Data represent the mean ± SE. * *p* < 0.05, ** *p* < 0.01. (**c**–**k**) Pearson correlation coefficient, r, was calculated for the expression level of CRBN (*y*-axis) and clinical parameters (*x*-axis) of 24 subjects.

**Figure 2 antioxidants-11-01980-f002:**
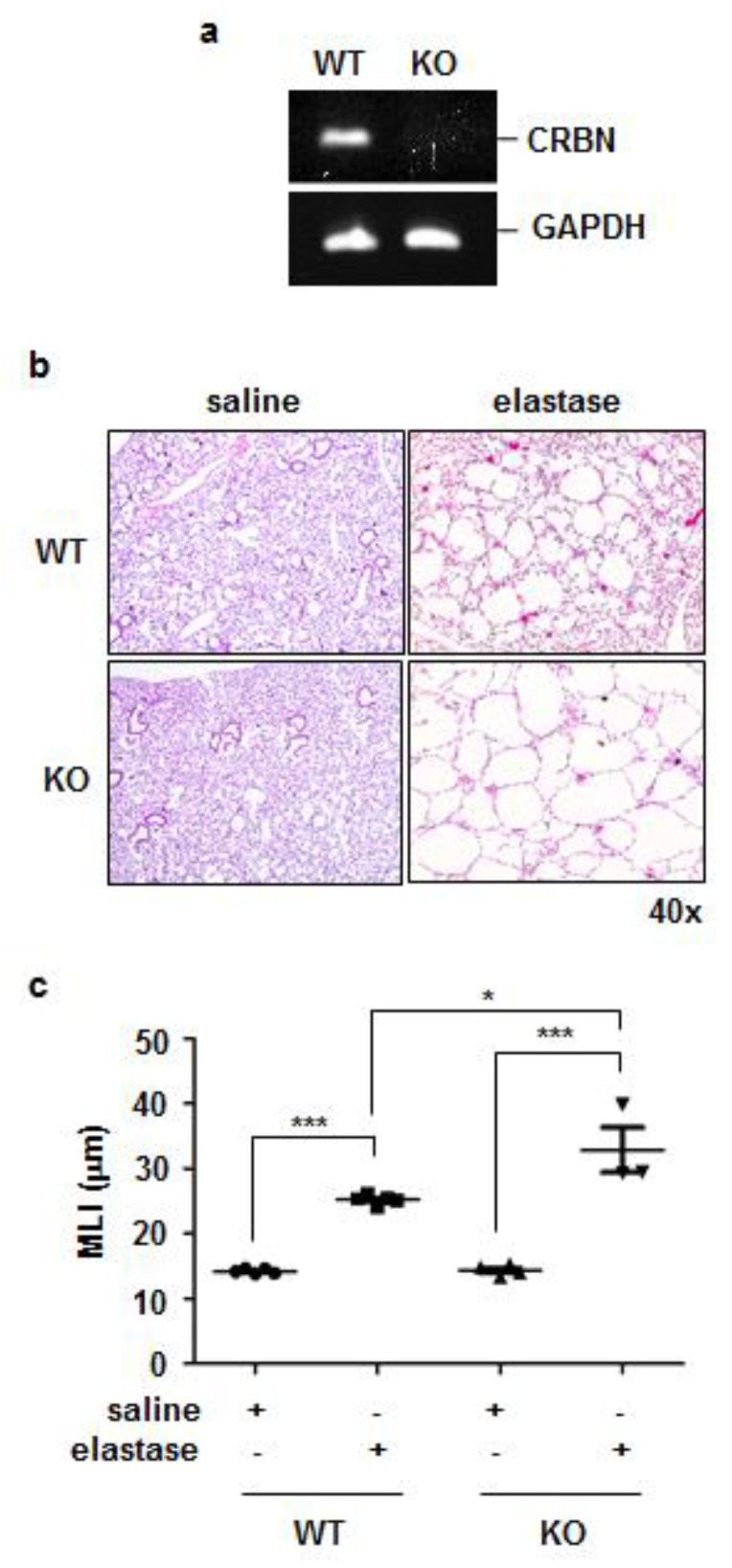
*Crbn* KO exaggerated elastase-induced emphysema in mice. WT and *Crbn* KO mice were given 0.5 units of elastase via intratracheal injection. (Day 14: WT-saline n = 5, WT-elastase n = 5, KO-saline n = 4, KO-elastase n = 3) (**a**) RT-PCR for *Crbn* and *Gapdh* in the lung tissues from WT and *Crbn* KO BALB/c mice. (**b**) Effect of *Crbn* deficiency on elastase-induced emphysema formation. Representative images of H&E staining in lungs from WT and *Crbn* KO mice 14 days after intratracheal instillation of elastase (×40). (**c**) The MLI was measured. Data represent means ± SE. * *p* < 0.05, and *** *p* < 0.001.

**Figure 3 antioxidants-11-01980-f003:**
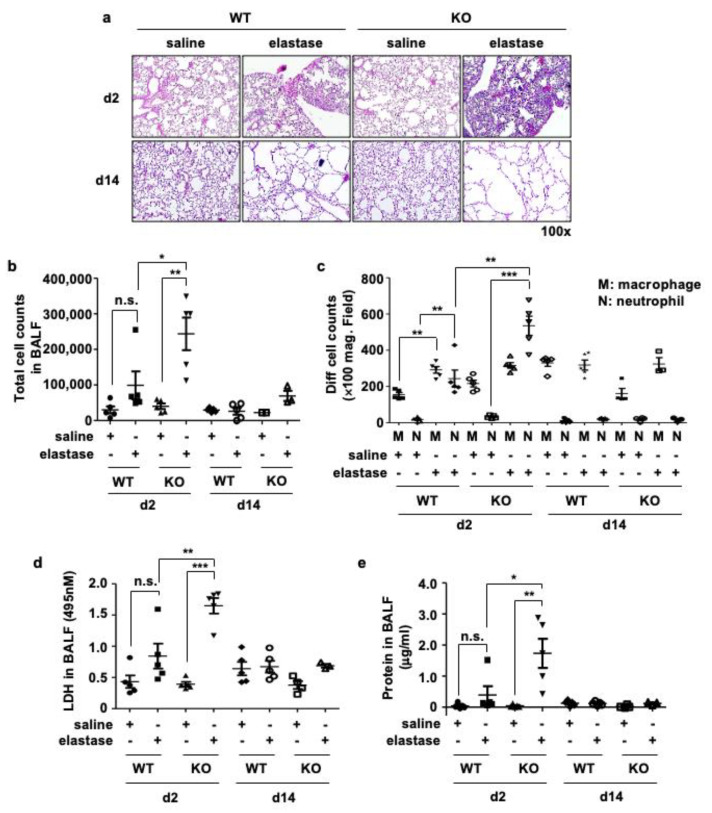
*Crbn* KO increased elastase-induced inflammation and cellular injury. WT and *Crbn* KO mice were given 0.5 units of elastase via intratracheal injection. (Day 2: WT-saline n = 5, WT-elastase n = 5, KO-saline n = 5, KO-elastase n = 5, Day 14: WT-saline n = 5, WT-elastase n = 5, KO-saline n = 4, KO-elastase n = 3). (**a**) Representative images of H&E staining in lungs from WT and *Crbn* KO mice on day 2 (d2) and day 14 (d14) after intratracheal instillation of elastase (×100). (**b**,**c**) BALF was collected on days 2 and 14 after intratracheal instillation of elastase. Total cells (**b**) and differential cells (**c**) including macrophages and neutrophils in BALF were counted. Data represent the mean ± SE. The concentrations of LDH (**d**) and protein (**e**) in BALF were measured. (**f**–**i**) Real-time PCR analysis of KC, TNF-α, IL-6, IL-1β, and GAPDH expression in the lung tissues of WT and *Crbn* KO mice. (**j**-**m**) The concentrations of KC, TNF-α, IL-6, and IL-1β were measured. Data represent the mean ± SE. * *p* < 0.05, ** *p* < 0.01, and *** *p* < 0.001, n.s., not significant.

**Figure 4 antioxidants-11-01980-f004:**
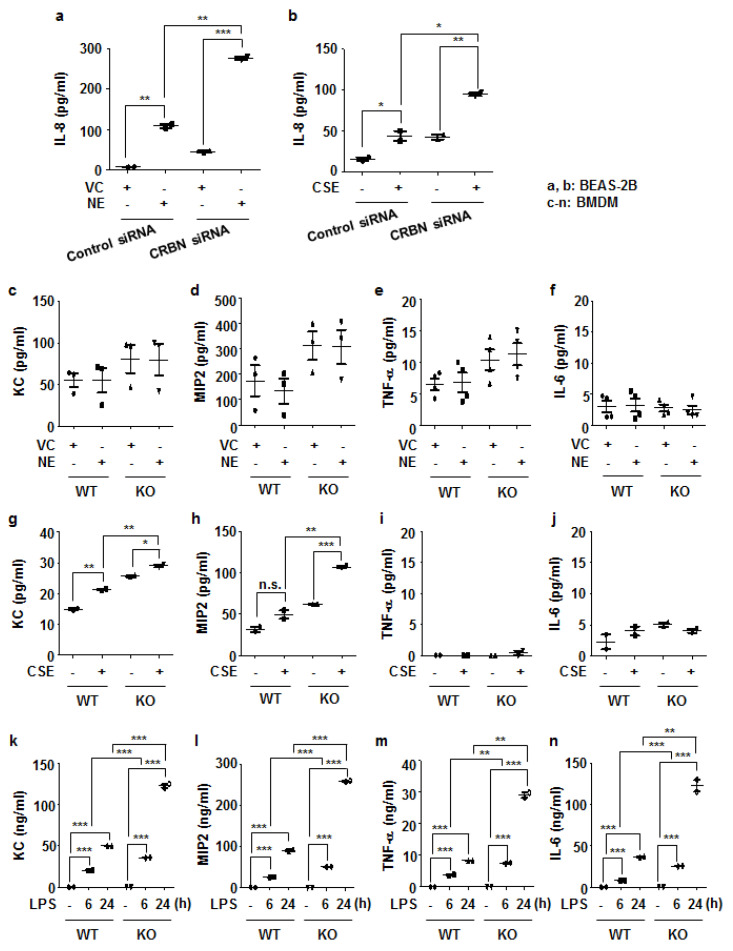
*Crbn* KO enhanced the production of inflammatory cytokines/chemokines in lung epithelial cells and macrophages. (**a**,**b**) BEAS-2B cells were transiently transfected with control siRNA or *Crbn* siRNA. Forty-eight hours after transfection, the cells were treated with VC or NE (1 U/mL) (**a**) and CSE (1%, *v*/*v*) (**b**) for 24 h. The concentration of IL-8 in cell supernatants was measured with an ELISA. Data represent the mean ± SD. * *p* < 0.05, ** *p* < 0.01, and *** *p* < 0.001 (**c**–**f**) WT and *Crbn* KO BMDMs were treated with VC or NE (1 U/mL) for 24 h. (**g**-**j**) WT and *Crbn* KO BMDMs were treated with CSE (1%) for 24 h. (**k**–**n**) WT and *Crbn* KO BMDMs were treated with LPS (100 ng/mL) for 6 or 24 h. The levels of inflammatory cytokines/chemokines in cell culture media were measured with an ELISA. Data represent the mean ± SD. * *p* < 0.05, ** *p* < 0.01, and *** *p* < 0.001.

**Figure 5 antioxidants-11-01980-f005:**
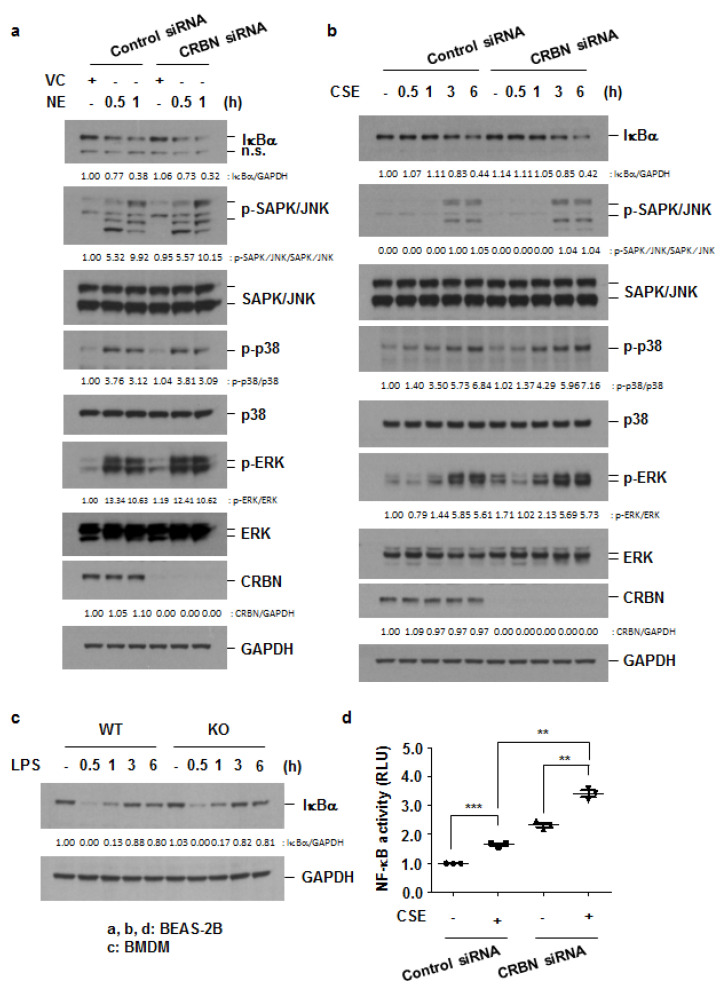
*Crbn* KO enhanced inflammatory cytokine/chemokine production by increasing NF-κB activation. (**a**,**b**) BEAS-2B cells were transiently transfected with control siRNA or *Crbn* siRNA. Forty-eight hours after transfection, the cells were treated with VC or NE (1 U/mL) (**a**) and CSE (1%) (**b**) for the indicated times. Total cell lysates were subjected to Western blot analysis for IκBα, p-SAPK/JNK, total SAPK/JNK, p-p38, total p38, p-ERK, total ERK, CRBN, and GAPDH. (**c**) WT and *Crbn* KO BMDMs were stimulated with LPS (100 ng/mL) for the indicated times. Total cell lysates were subjected to Western blot analysis for IκBα and GAPDH. (**d**) BEAS-2B cells were co-transfected with the NF-κB-luciferase reporter construct, control plasmid, control siRNA, and *Crbn* siRNA using the Neon electroporation kit. Forty-eight hours after transfection, cells were treated with CSE (1%) for 24 h. Luciferase activity was detected and normalized with Renilla activity. Data represent the mean ± SD. ** *p* < 0.01, and *** *p* < 0.001.

**Figure 6 antioxidants-11-01980-f006:**
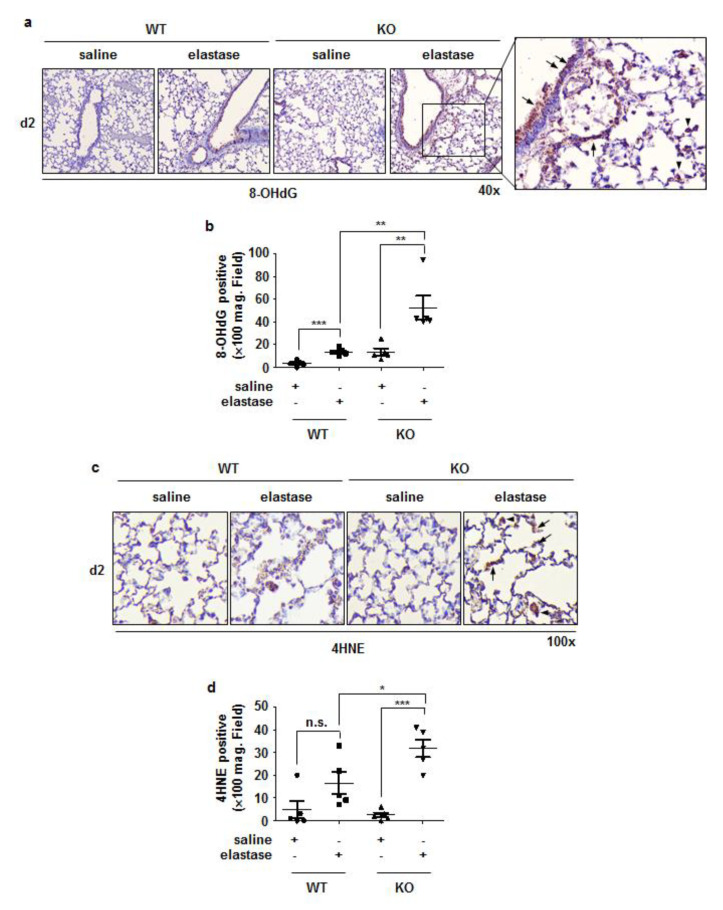
*Crbn* KO increased elastase-induced oxidative damage. WT and *Crbn* KO mice were given 0.5 units elastase via intratracheal injection. (Day 2: WT-saline n = 5, WT-elastase n = 5, KO-saline n = 5, KO-elastase n = 5). Representative images of 8-OHdG (**a**) and 4HNE (**c**) immunohistochemical staining in lungs from WT and *Crbn* KO mice on day 2 (d2) after intratracheal instillation of elastase. (**b**,**d**) Quantity of 8-OHdG positive cells and 4HNE positive cells in ×100 field of the whole group. Data represent the mean ± SE. * *p* < 0.05, ** *p* < 0.01, and *** *p* < 0.001, n.s., not significant.

**Figure 7 antioxidants-11-01980-f007:**
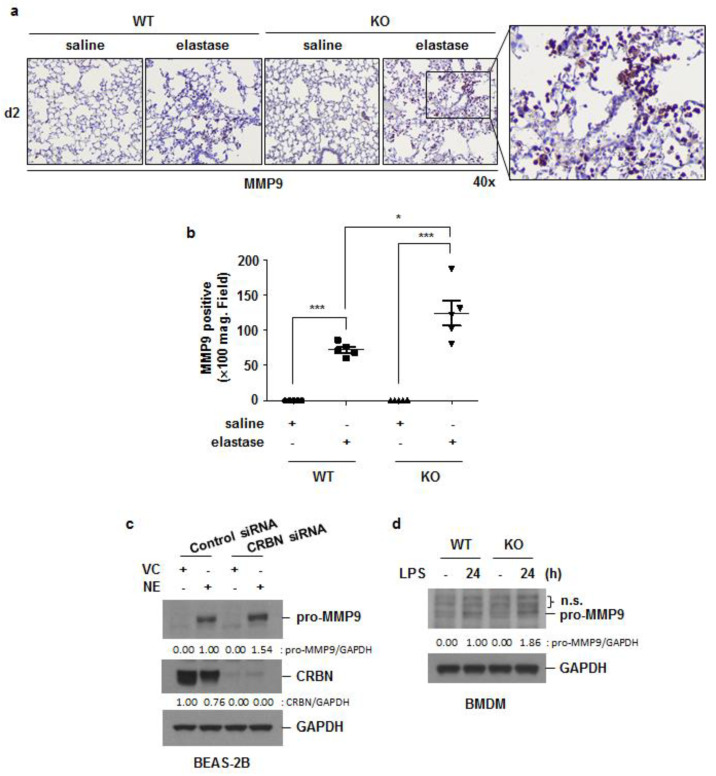
*Crbn* KO increased elastase-induced MMP9 expression. WT and *Crbn* KO mice were given 0.5 units of elastase via intratracheal injection. (Day 2: WT-saline n = 5, WT-elastase n = 5, KO-saline n = 5, KO-elastase n = 5). (**a**) Representative images of MMP9 immunohistochemical staining in lungs from WT and *Crbn* KO mice on day 2 (d2) after intratracheal instillation of elastase (×40). (**b**) Quantity of MMP9 positive cells in ×100 field of the whole group. Data represent the mean ± SE. * *p* < 0.05, and *** *p* < 0.001. (**c**) BEAS-2B cells were transiently transfected with control siRNA or *Crbn* siRNA. Forty-eight hours after transfection, the cells were treated with VC or NE (1 U/mL) for 24 h. (**d**) WT and *Crbn* KO BMDMs were treated with LPS (100 ng/mL) for 24 h. Total cell lysates were subjected to Western blot analysis for MMP9, CRBN, and GAPDH.

**Figure 8 antioxidants-11-01980-f008:**
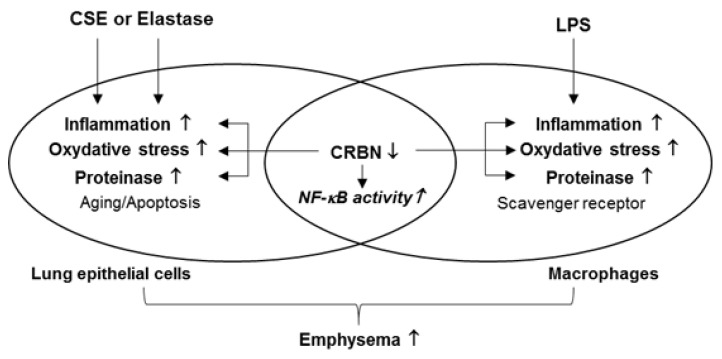
Proposed mechanism for the role of CRBN on the development of emphysema. *Crbn* deficiency enhances elastase-, CSE-, or LPS-induced inflammation, oxidative damage, and MMP9 expression in lung epithelial cells and macrophages. However, cellular aging and apoptosis in lung epithelial cells and the surface expression levels of scavenger receptors such as MARCO and SR-A in macrophages were not affected by CRBN expression. Increased inflammation, oxidative damage, and MMP9 expression induced by *Crbn* deficiency might augment the development of emphysema.

## Data Availability

The data presented in this study are available within the article. Other data that support the findings of this study are available upon request to the corresponding authors.

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
