# Peer review of "Cereblon Deficiency Contributes to the Development of Elastase-Induced Emphysema by Enhancing NF-κB Activation"

_antioxidants, 2022, doi:10.3390/antiox11101980_

Round 1
Reviewer 1 Report
Comments for authors:
The authors for this paper investigate role of cereblon (CRBN) in the pathogenesis of the chronic airways disease COPD. CRBN has been shown to play a role in regulating inflammatory responses in other diseases, so authors wanted to determine if it has a similar role in COPD. Using a CRBN-/- mouse model, authors show here that a lack of CRBN induced some features of disease, including airway neutrophilia, worsened emphysema, signs of lung oxidative damage and remodelling.
General Comments
This is an interesting study that highlights a potentially new therapeutic target for COPD treatment. This is a solid body of work, but there are a few areas that would benefit from additional data/analysis and consideration for discussion.
Specific comments/questions:
Results:
In general, the graphs are too small, and all the titles and labels on the axes are too small to comfortably read. Please make all of these bigger.
It’s also hard to follow the data when BMDM and BEAS-2B data are mixed interchangeably in a single figure. Either consider putting all the BMDM data together and all the BEAS-2B data together, or if that doesn’t work, then clearly labelling each part of the figure so that it’s obvious what each graph is.
Use of standard statistical nomenclature on the graphs would be preferable; i.e. *= p<0.05, ** = p<0.01 and ***= p<0.001, instead of using ** to represent all statistical data.
Figure 2 - Why were there not 5 mice in the KO-saline and KO-elastase groups?
· Figure 4 – It would have been better to have investigated inflammatory responses in the NE-treated CRBN-/- mice. Authors would have lung tissue to do qPCR and/or western blots for chemokines/cytokines in the lungs. If authors were interested in the role of CRBN on host responses to LPS, mice could have also been treated with LPS.
· Figure 5 – There is a lot of western blot data that needs to be quantified, representative blots are not enough to determine changes in expression.
· Figure 6 – Part B, WT elastase panel is very blurry and needs to be replaced with a clearer picture.
Discussion:
Inflammation is an important component of COPD, however the authors don’t show any induction of common cytokines in their wildtype CSE and NE-treated BMDM models (except for neutrophil chemokines/cytokines). I would expect some of these cytokines to be expressed. Were these models optimised? Can the authors discuss why they don’t see any IL-6, TNF-alpha or other cytokines induced by their treatments? This may also influence how they interpret the results of the CRBN-/- BMDM data.
On line 363 of the discussion, authors say that their data “suggest that patients with reduced levels of CRBN could be more susceptible to COPD exacerbations”. To make this conclusion, the authors would need to have investigated the effect of LPS on either CSe or NE-treated cells, so this is a somewhat pre-mature statement.
Authors also don’t discuss any of the human data in figure 1 and how this relates to the rest of their work, this is needed.
Author Response
Please see the attachement.

Reviewer 2 Report
Heo et al. evaluated the role of cereblon in the mechanism of elastase-associated emphysema. They found that deficiency of cereblon worsens emphysematous changes and infiltration of inflammatory cells in the lungs. They also found decreased expression of cereblon in the lungs of patients with COPD and increased expression of inflammatory mediators in cells derived from mice deficient in cereblon. The study was relatively well conducted, and the story was clear. However, there are some questions and corrections that need to be made.
1. It would be important to display the data in a dot plot in all figures to confirm the distribution of the data.
2. The statistical analysis should be described in each figure. The authors described that they used ANOVA. They should describe what type of ANOVA and the post hoc analysis used to evaluate the difference between groups.
They also evaluated correlations in the human samples. The statistical method they used for evaluating correlations between the parameters should be described in the method section.
3. It would be very informative to show the levels of some cytokines (MCP-1, TNF-alpha, IL-1b, etc.) in the BALF of the mouse models.
4. In figure 6, the markers of oxidative stress should be quantified.
5. In figure 7, the lung staining in a and the Western blotting should be quantified.
6. The authors should describe the limitations of the study and future connotations.
Round 2
Reviewer 2 Report
No more comments